# The Production and Delivery of Probiotics: A Review of a Practical Approach

**DOI:** 10.3390/microorganisms7030083

**Published:** 2019-03-17

**Authors:** Kurt Fenster, Barbara Freeburg, Chris Hollard, Connie Wong, Rune Rønhave Laursen, Arthur C. Ouwehand

**Affiliations:** 1DuPont Nutrition and Health, Madison, WI 53716, USA; kurt.fenster@dupont.com (K.F.); barbara.freeburg@dupont.com (B.F.); chris.hollard@dupont.com (C.H.); Connie.wong@dupont.com (C.W.); 2DuPont Nutrition Biosciences ApS, 8220 Brabrand, Denmark; runerl@gmail.com; 3DuPont Nutrition and Health, 02460 Kantvik, Finland

**Keywords:** probiotics, *Lactobacillus*, *Bifidobacterium*, manufacturing, strain stability

## Abstract

To successfully deliver probiotic benefits to the consumer, several criteria must be met. Here, we discuss the often-forgotten challenges in manufacturing the strains and incorporating them in consumer products that provide the required dose at the end of shelf life. For manufacturing, an intricate production process is required that ensures both high yield and stability and must also be able to meet requirements such as the absence of specific allergens, which precludes some obvious culture media ingredients. Reproducibility is important to ensure constant high performance and quality. To ensure this, quality control throughout the whole production process, from raw materials to the final product, is essential, as is the documentation of this quality control. Consumer product formulation requires extensive skill and experience. Traditionally, probiotic lactic acid bacteria and bifidobacteria have been incorporated in fermented dairy products, with limited shelf life and refrigerated storage. Currently, probiotics may be incorporated in dietary supplements and other “dry” food matrices which are expected to have up to 24 months of stability at ambient temperature and humidity. With the right choice of production process, product formulation, and strains, high-quality probiotics can be successfully included in a wide variety of delivery formats to suit consumer requirements.

## 1. Introduction

The most widely accepted definition of probiotics is the one proposed by a working group of the FAO/WHO in 2002 [1] and confirmed with minor grammatical changes by an ISAPP expert panel [2]: “Probiotics are live microorganisms that, when administered in adequate amounts, confer a health benefit on the host”. This definition implies five important things (Table 1).

In the present article we will focus on points 2–4 in Table 1; the applied side of using probiotics. Coming the type of microorganisms, we will be focusing on lactic acid bacteria (LAB) and bifidobacteria; the most commonly used probiotic genera. However, species from other genera, such as e.g., Bacillus and Saccharomyces, are also used as probiotics. Organisms from these genera may have very different growth requirements and stability properties [3]. As viability is key, at sufficient amounts and until end of shelf life, probiotics need to be manufactured in such a way that they are robust and stable. They also need to be included in consumer products that allow their survival, in sufficient numbers, until end of shelf-life.

The definition does not stipulate what an adequate amount is. However, regulators in, for example, Canada and Italy require a minimum dose of 10^9^ colony forming units (CFU) [2]. Furthermore, the adequate amount can be assumed to be at least the dose that was documented to provide the specific health benefit to which is referred. There is no indication that a higher dose is detrimental [4,5]; in some cases, it may be beneficial [6]. Probiotics, particularly when included in dietary supplements, are commonly transported and stored at ambient temperatures and humidity. This may lead to loss of viability as compared to refrigerated/frozen storage and handling. In order to provide the target dose until end of shelf life and to compensate for potential losses during storage and handling, an overage is commonly included in the product [7].

There are many articles available on the identification of potential new probiotics and their safety [8], as well as about the health benefits of specific probiotic strains or strain combinations [9]; however, these topics are not discussed here. Also, for the market potential and health-economics of probiotics, the reader is referred to elsewhere (e.g., [10,11]). Here, we discuss what is required to reliably and reproducibly produce high-quality, safe, and stable probiotics and what it takes to keep them alive in sufficient numbers in various delivery formats in order to provide efficacious probiotics and their health benefits to the consumer.

## 2. Manufacturing Probiotics and Dairy Starter Cultures

LAB and bifidobacteria are commercially manufactured to satisfy customer demand for probiotic dietary supplements and dairy starter cultures. From a manufacturing standpoint, the desired commercial product will have as high a yield as possible and consist of viable, concentrated cells that are stable and will have consistent performance in the intended application. High cell count and long shelf-life stability in a variety of different temperature and humidity conditions are expected by customers, especially for high-quality dietary supplement products with doses established through clinical trials. In contrast, rapid and consistent acidifying activity in milk is desired by customers for dairy starter cultures. In this section, the manufacturing process is briefly described and important challenges are highlighted for manufacturing and consistent product performance.

The manufacturing processes of LAB and bifidobacteria for dietary supplements and dairy applications have the following steps in common, as shown in Figure 1. Frozen seed stock, which has been carefully prepared to consist of a single pure strain and verified to be free of contaminants by quality control (QC) testing, is used in a limited number of sequential seed fermentations to achieve the desired inoculum volume and is ultimately transferred to the main fermentation vessel for growth. Alternatively, frozen direct vat inoculation (DVI) material consists of a larger amount of concentrated cells that can be used to directly inoculate the main fermentation vessel. The aim of both approaches is to limit the number of generations from seed stock to product, thereby reducing any potential risk for genetic drift. The heat-treated medium used in the seed scale up and main fermentation is a blend of water, nitrogen sources, carbohydrates, salts, and micronutrients necessary for growth. The fermentations are carefully controlled and after the fermentation in the main tank is completed, the cells are concentrated by separating the cells from spent medium through centrifugation. Depending upon the final product application, stabilizer solutions (i.e., cryoprotectants to protect cells from injury during freezing and/or lyoprotectants to protect cells from injury during freeze-drying) may be added to the cells prior to freezing. Cryoprotectants inhibit the rate of ice growth via increasing the solution viscosity and keeping the amorphous structure of ice in close proximity of the cell. Lyoprotectants stabilize the lipid bilayer structure of the cell membrane in the absence of water [12]. Commonly used cryo- and lyoprotectants are carbohydrates and peptides. In the dairy industry, skim milk powder is often used [13]. Once the probiotic concentrate is blended with the cryoprotectant solution, various freezing processes can be applied. One simple freezing technique consists of pouring cryoprotected concentrate into cans and immersing the sealed cans into a liquid nitrogen bath. The frozen cans can then be shipped to companies incorporating probiotics in food or beverages. Alternatively, a more productive technique consists of pelletizing the cryoprotected concentrate by dripping the concentrate through calibrated holes into a bath of liquid nitrogen. The pellets, which are typically spheres of 4–5 mm in diameter, are then harvested at the bottom and finally packed into bags that are stored and shipped at a temperature ranging from −45 to −55 °C. Alternatively, frozen cell pellets can be used for freeze-drying (lyophilization) to a dried end-product. The frozen pellets are transferred onto trays which are placed on top of shelves. The shelves have the capability of being temperature controlled and are progressively heated once vacuum is established in the freeze-drying chamber. An alternative option consists of filling trays with the cryoprotected concentrate. The trays are then placed on top of temperature-controlled shelves which are initially cooled down to freezing temperatures under atmospheric pressure. Once the concentrate in each tray is frozen, the shelves are gradually heated once vacuum is applied. The applied vacuum typically varies between 100 and 1000 mTorr and the shelves’ temperature between −40 and +40 °C. Freeze-drying length varies as a function of the strain, its formulation, and the freeze-drying cycle but usually takes a few days to be completed. The advantage of freeze-drying is that the process maintains the probiotic cells at a low temperature to limit damage to the cells’ structure and metabolites [14].

After removal from the dryer, the lyophilized material is milled to a powder with a defined particle size and density. The milled material can then be used for blending with excipients (bulking agents), additional functional ingredients if required, and flow aids, depending on the needs of the customer. The blend is then used to make finished formats such as capsules, sachets, or tablets. QC testing is performed on in-process samples and the final product to make sure that the end-product is high quality and free of contamination.

## 3. Development of Strain Production

During development work, special care is taken to understand the production conditions involved in manufacturing probiotics at a commercial scale and to evaluate the performance of strains under similar conditions at lab scale. Each step in the process depends upon the prior step and it is important to identify strain-dependent sensitivities and to try to maintain the overall health of the cells while proceeding through the process. Scale up can sometimes be very challenging because the down-sized process for making cells during lab-scale development work is inherently more tightly controlled and has shorter hold times during each step in the process. For example, commercial separation of cells from spent media by centrifugation may take hours because of the large volume of cells compared to minutes at lab scale with the smaller volumes involved; this leads to different stresses (i.e., usually heat and sheer stress) than those encountered using an upright lab-scale centrifuge [15]. Also, there are multiple steps where cells are pumped during commercial-scale production which do not typically happen during bench-scale development work. In addition, cells during commercial production are likely to experience different pH and temperature conditions that are not easily reproduced in the same way at lab scale. Hence, the importance of scaling up to an intermediate volume in pilot, so that these more representative production conditions and stresses can be evaluated and mitigated before proceeding to commercial production. Scale up from pilot to commercial scale maybe challenging for the same reasons as scaling up from lab scale to pilot.

As mentioned previously, the hold times at various steps in the production process can greatly exceed those at lab scale. In order to ensure that the cells are robust and will meet shelf-life claims, the cells manufactured in pilot- and commercial-scale production are evaluated for several hours past the typical hold time that they would normally encounter at the different steps in the process. If adequate robustness is not demonstrated and an adjustment cannot be easily made to mitigate this sensitivity, the strain will return to the laboratory for another iteration of development work before being scaled up again.

Generally, if a strain seems to be especially sensitive and difficult to develop at lab scale because of strain-dependent sensitivities, it is highly likely that additional challenges will be encountered during scale up to pilot and subsequent scale up to commercial production. The same is true for the robustness of the cells with the hold times involved in the production process. Often, there are additional iterations where the strain will return to the laboratory for additional development work to try to redevelop and overcome these identified sensitivities and robustness issues before scaling up again.

Like fermentation, freeze-drying must be evaluated at the bench-scale level before commercial production. An optimal freeze-drying cycle can be defined through an iterative process where pressure, heating plate temperatures, and frozen pellet bed thickness are adjusted until appropriate water activity (Aw), cell count, and shelf-life stability are achieved [16]. In addition, cryoprotectant formula or dosage can be adjusted if cell survival is not satisfactory after the iterative process [17]. It is imperative to determine if the lab freeze-drier operation can be scaled up to an industrial freeze drier. In particular, the condenser capacity, condensing rate, and heat transfer of the industrial freeze drier must be known and sufficient to eliminate the moisture released during the drying cycle.

## 4. Strain Nutritional Requirements

LAB and bifidobacteria are fastidious microorganisms in terms of nutritional requirements for growth and performance. LAB and bifidobacteria tend to be auxotrophic for some of the 20 amino acids and have nutrient requirements that need to be satisfied from the external environment to grow. The complexity of these auxotrophies and nutrient requirements is often linked to the nutritional potential of the environment to which the microorganism was adapted and from which it was sourced [18]. For example, *Lactobacillus plantarum* sourced from plant material has fewer auxotrophies [19] and more biosynthetic self-sufficiency than *Lactobacillus johnsonii* isolated from the upper gastrointestinal tract of a human being, which is an environment with a greater availability of nutrients such as free amino acids, short peptides, and oligosaccharides [20]. Understanding the microbe’s nutritional requirements and developing a tailored fermentation medium that supports growth and enhances the ability of the cells to survive and adjust to the stresses imposed by the manufacturing process is key to having a high-performance end-product.

Identifying strain-dependent nutritional needs requires a multidisciplinary approach that is both knowledge and empirically based. The power of evaluating the genome (i.e., genomics), gene expression (i.e., transcriptomics), protein expression (i.e., proteomics), and metabolism (i.e., metabolomics) provides critical knowledge of strains that is useful for assessing strain-dependent nutritional needs and capabilities and contributes to development work and the ultimate performance of the manufactured product [21,22,23]. Also, analyzing the sterilized medium prior to inoculation and after fermentation (i.e., spent medium) provides empirical results for nutritional needs and limitations that can be used to corroborate and crosscheck more knowledge-based information derived using -omics. Using these approaches to also understand the composition of complex raw ingredients, yeast extracts, yeast peptones, milk, and other complex nitrogen sources helps match critical fermentation media ingredients with the nutritional needs of the strain under development and provides an opportunity to adjust the medium and process to achieve better strain performance and better manage manufacturing costs and efficiency. Finally, there is a wealth of empirical data that can be collected that is not easily obtained or even predicted using the more knowledge-based -omics approaches. With the right expertise and innovative philosophy, these approaches, when combined, are extremely powerful for understanding strain dependencies, strain sensitivities, nutritional needs, and nutritional limitations, so that high-performance strains can be successfully developed and manufactured that satisfy customer needs.

## 5. Manufacturing Raw Materials

Given the importance of the fermentation medium for manufacturing LAB and bifidobacteria, changes to the raw materials can have a profound effect on growth and performance. The changes to raw materials by the supplier could be due to cost savings with process improvements, a change in ingredient sourcing, and variation within the manufacturing process. Such changes to complex ingredients such as protein sources (e.g., yeast extract, milk) are not surprisingly more often pronounced in terms of compositional differences than less complex ingredients such as simple carbohydrates and salts. Depending on the nutritional requirements and sensitivities of the strains being manufactured, the lot-to-lot variation in complex raw ingredients can sometimes go undetected, with some strains having seemingly consistent performance, whereas the performance of other strains is more obviously affected in a positive or negative way. With more complex ingredients such as yeast extracts, the differences responsible for the change in strain performance may not be readily attributed to the amino acid, peptide size distribution, vitamin, nucleotide, salt, and carbohydrate levels but rather due to the presence or absence of some other unknown or less obvious components. Beet and cane molasses are used to grow baker’s yeast that will be used for the production of yeast extracts and peptones for food applications and fermentations [24]. Carryover of components used to culture the yeast to make the yeast extracts and peptones can have no obvious effect on strain performance or can positively or negatively affect the performance of probiotic strains in a strain-dependent manner. Also, cane and beet molasses can be sourced from all over the world with performance and quality which can have lasting effects when carried over into fermentations with yeast extracts and peptones [25].

## 6. Process Control and Consistency

Manufacturing LAB and bifidobacteria with consistently high performance is also dependent upon how well controlled the manufacturing process is. Unsurprisingly, there is considerable diversity between strains, even from the same species, in terms of sensitivity and response to the manufacturing process, which affects performance [26,27]. During lab-scale development, scale up to pilot, and subsequent commercial scale up, these sensitivities are discovered and the process is adjusted so that consistent high performance can be achieved. Once the process for each strain is established, it is important to run this process the same way each time. Controlling the manufacturing process occurs at several levels that include the following:The raw material suppliers are audited and raw materials evaluated at some level to ensure high quality.Establishing meaningful and achievable ranges for process parameters and verifying process capability for consistently being within those ranges.Automating the process as much as possible to reduce inconsistency associated with human error and manually controlled aspects of the manufacturing process.Making sure that operators are adequately trained and that employee turnover is low.Evaluating captured data from the process and using Six Sigma approaches [28] for continuous improvement and making sure the process is being reproduced as consistently as possible. Evaluation of in-process samples and final product to ensure the product is of high performance and free of contaminants.

The reality of the manufacturing environment is that there will be lot-to-lot variation in raw materials that will not be discernible until they are used in manufacturing. The production plant environment is dynamic, with new equipment installations and new processes being implemented that can perturb plant steady-state operation for a time. Also, some aspects of the manufacturing process are going to be more manual and less automated. There will be shift changes for operators and employee turnover. Equipment can and will unexpectedly fail. Probes and sensors used to monitor different steps in the manufacturing process can malfunction. Each of these examples are challenges that can affect the performance of the strains being manufactured, because in these circumstances, the process conditions and hold times are different and perhaps outside the range explored during strain development work and scale up. Manufacturing experience has demonstrated that process differences resulting from examples such as these, even seemingly minor and superficially unimportant, can have an outsized impact on performance that is usually, but not always, negative for performance. Even strains considered to be well understood and reliably manufactured can have surprisingly poor performance if the change or difference in the process is outside what was established during development, scale up, and implementation. Sometimes, efforts to troubleshoot and correct the process issue is confounded because the differences between production runs is multivariate and not readily identifiable. This suggests that some aspects of the process are not adequately monitored and controlled to ensure consistently high strain performance. The importance of controlling the process cannot be understated for consistently manufacturing high-performance LAB and bifidobacteria, especially given the strain-specific nutritional requirements and process sensitivities, as well as subsequent cell responses to the process steps that affect performance.

## 7. Species and Strain Dependencies

In order to manufacture high-performance probiotics and dairy starters, the unique nutritional requirements and sensitivities of different aspects of the manufacturing process of each strain have to be well understood and accommodated within the manufacturing process. These strain dependencies are typically identified and worked out during lab-scale development, scaled up in pilot, and scaled to the commercial level before the process is finalized and commercial manufacturing proceeds. Developing a tailor-made manufacturing process to accommodate these strain dependencies poses additional challenges because of the complexity associated with sourcing and managing raw materials, as well as managing numerous manufacturing processes in the production facility. Fortunately, the strain dependencies associated with mesophilic (e.g., *Lactococcus lactis*) and thermophilic (e.g., *Streptococcus thermophilus*) dairy starters are less pronounced, and more shared fermentation media and manufacturing processes with these microorganisms are possible. Conversely, the complexity of the manufacturing processes for probiotics destined for dietary supplements is considerably greater because of the ever-increasing number of probiotic species in demand by customers and the diversity of species and strain dependencies that are involved.

To successfully manufacture high-performance probiotics and dairy starters, the strains need to be well understood in terms of nutritional needs [18,19] and process sensitivities [8]. Also, the composition of complex raw materials (i.e., yeast extracts) and how the nutritional value changes leading up to and including the main fermentation needs to be well understood. The manufacturing process has to be very well controlled, so the strain can be consistently made with subsequent predictable performance. Finally, the organization has to be willing and able to manage the high degree of complexity incurred through the ever-increasing number of raw materials and tailor-made processes necessary for manufacturing high-performance dairy starters and probiotics.

## 8. Raw Materials for Growth Media with Special Requirements

As outlined above, raw materials for the production of probiotics and dairy starter cultures need to be carefully selected and controlled. However, in addition to the growth requirements dictated by the organisms, there may be requirements for the media from the customer and/or consumer. These may relate to Kosher and Halal requirements but may also concern the absence of certain allergens from the final product. This maybe challenging, as some of the commonly used media for the culturing of microorganisms in general and LAB and bifidobacteria in particular would not comply with these requirements. For Kosher, this means Kosher ingredients, no mixing of dairy and meat products, and the use of Kosher methods. For allergens, avoiding the most common dietary allergens, such as dairy, soy, gluten, and nuts. For vegetarian and vegan production, meat and dairy sources have to be avoided, respectively. For dairy starter cultures, obviously a dairy-based medium can be used. For dietary supplements, the omission of dairy, soy, and meat extracts requires searching for alternative nitrogen sources. Carbon sources derived from wheat have to be avoided because of potential contamination with gluten. As is outlined below, it is important that this is documented and checked, and if necessary, analyzed, in order to be able to guarantee the absence of specific allergens and to guarantee Kosher, Halal, vegetarian, and/or vegan products.

## 9. Evolution of Quality Control

Quality control and quality assurance both have the same goal of producing a quality product for sale, but they differ in their approach. Quality assurance is responsible for maintaining quality systems within the facility so that product defects and mistakes can be kept to a minimum. Quality control is responsible for the actual testing of raw materials, in-process samples, intermediate samples, and end-product samples, which can involve a wide variety of examinations. The need for end-product testing has always been QC’s primary task, but what has changed and expanded are the QC support programs. Before end-product testing can occur and the results considered sufficiently reliable to be added to a certificate of analysis (COA), the QC lab needs to have implemented several programs which help to ensure the quality of the product.

As ever-more probiotics are being consumed by the general population, which includes vulnerable subjects such as pregnant women, infants, people with allergic reactions, and those with compromised immune systems, the standards and rules for maintaining QC labs and testing have advanced [29]. Compliance with regulatory guidance and applicable standards have improved the control and outcome of production and QC processes, augmenting the ability of production facilities to produce and QC to release a more consistent end-product. Also, by implementing and practicing such guidelines, continued areas for development and/or corrections can be adopted more readily. Examples of regulatory guidance and applicable standards that are being embraced are given in Table 2.

Additionally, by adopting good laboratory practices (GLP), the QC lab minimizes cross contamination by understanding the steps used in production to make the end-product, the composition of the end-product, and the handling of the end-product. This includes personal hygiene, the proper personal protective equipment (PPE), establishing foot traffic protocols, product flow through the lab, and sanitation procedures and logs. Qualification of equipment and validated procedures/methods are also utilized so that the test results are reliable. Hazard analysis and critical control points (HACCP) is used to identify critical control points and establish acceptable protocols to minimize hazards. Utilizing customer as well as in-house auditing of the QC lab attempts to identify continued areas of concern and to make sure that methods are being followed.

Metrology is equally as important as GLP in establishing a robust program for the qualification and calibration of lab equipment. Programs need to be set in place to make sure that the equipment, such as autoclaves, incubators, clean room environments, pipettes, etc., is sufficiently monitored and maintained both by qualified outside vendors and, on a more daily occurrence, the lab technicians themselves. Metrology records are also created, maintained, and retained. Also included is the monitoring of the air handling and water quality, which is important in preventing any contamination downstream.

Documentation needs to include all observations, test results/raw data, and deviations. Every lot of material which is produced should be associated with a batch record which includes all production and QC paperwork. That file should then be reviewed by quality assurance to make sure all relevant paperwork is present and regulatory compliance is met. The length of time which all batch records, metrology records, etc., should be kept and how to properly dispose of all company paperwork depends on the regulatory guidance practiced and company policy.

To avoid quality issues being caused by inferior raw materials and packaging, vendors and their raw materials should be qualified for use ahead of time. This means that the following information should be known and approved by the QC team before use:GMO status; allergen status; the raw material purchasing specifications, including chemical, physical, and microbiological; food grade quality; pesticides; irradiation; Kosher rating; raw material packaging type and size; storage condition; shelf life; and the safety data sheet (SDS) should be reviewed. It is also helpful to review a COA from the vendor.It should also be determined what type of inspection needs to occur once an approved raw material arrives at the plant. This can include appearance; identification; chemical, physical, and microbiological; review of the vendor COA; identification that the material is proven to be what it is; and the frequency of the QC checks. For example, does a QC of the raw and packaging materials need to happen with every new batch, or can a skip program be established?

The trained skills of the laboratory technicians should be monitored by establishing a program whereby the accuracy of quality tests can be checked against known controls. Retaining skilled laboratory technicians and maintaining their skills will help minimize inconsistency among results.

A sampling plan will need to be developed which encompasses raw/packaging materials, in‑process/intermediates, and end-products. Information which needs to be considered in order to create a representative sampling plan is: how many samples should be taken which accurately represent the batch? Statistical programs have been developed, which take into consideration the sample(s), that provide the reassurance that the batch is acceptable while avoiding over-testing, which saves both technician time and money. Additionally, if more than one sample is taken, or if production runs multiple sublots within a batch, do they need to be run separately or can samples be consolidated? Sample size also needs to be considered.

Retained samples from each production lot should be stored at the same recommended temperature supplied to the customer. The sample size should reflect material taken over the production run and accommodate full QC testing.

There are also programs set up which involve product returns/customer complaints/deviations/out of specification and scrap. These are all mandatory programs which help prevent unintended use of the product and/or material and in which QC plays a role. In general, all of these events involve the steps indicated in Table 3.

The creation of release specifications needs to include customer requests, the capability of production, and the absence of pathogens.

With the support systems in place, questions concerning the type of end-product testing required need to be addressed. The end-products are bacteria, but those bacteria do not undergo any further fermentation in the customer’s end-product. Instead, the customer directly ingests the bacteria. How are the bacteria to be used in the customer’s end-product? What type of environment surrounds the bacteria to be sold to the consumer? That is, are the bacteria in a freeze-dried form that will be packaged in capsules, sachets, straws, etc., or will the bacteria remain in a frozen wet pellet form that can be added to liquids such as juice? What is the acidic/basic conditions of the customer’s end-product and how will it affect the stability of the bacteria? What type of consumer is being targeted by the customer? Infant? Senior citizen? Immunocompromised? Gender? Examples of end-product testing are listed in Table 4.

Future concerns are efficiency and automation. A need for future methods that can be validated and accepted by the customer is in definite need. The QC lab, along with help from the R&D department, is always looking for ways to reduce the release time and prevent any repeat testing. Additionally, customers want more accurate identification, especially among bacterial blends.

## 10. Inclusion of Probiotics in Dietary Supplements

The inclusion of probiotics in dietary supplements primarily utilizes probiotics in the freeze-dried powder format. Capsules, tablets, and powder in stick packaging or sachets are the most commonly found formats on store shelves and are usually stored at ambient conditions. Dietary supplement products should deliver the probiotic count declared on the label throughout the shelf life of the product. This ensures that the consumer receives the adequate dose of probiotics to affect the targeted structure function health claim or the otherwise suggested health benefit. It is important to characterize the stability of each strain so that the proper amount of overage can be added during the production of the dietary supplement format to ensure minimal counts of each component strain in a multi-strain formulation. However, the quality control of single strains in a multi-strain formulation is very challenging, especially with multiple strains from the same species. Currently, no generally applicable and reliable methods exist, although experimental molecular-based techniques have been explored [30].

Dietary supplement formats typically have shelf lives measured in years, so a great deal of care must be taken to make sure that the proper probiotic cell count is maintained. Because probiotics, even in the freeze-dried state, are live microorganisms, more consideration must be given to their handling and storage than for other dietary supplements and food ingredients [16,31].

Water activity, followed closely by storage temperature, is the main factor impacting probiotic stability over the shelf life of the product [32]. Downstream processing and manufacturing of the final product (dietary supplement) needs to happen under strict temperature- and humidity-controlled conditions. Establishing a low-water-activity product starts with sourcing dry carriers, excipients, and other active ingredients that will be blended with the probiotic. Small amounts of high-water-activity ingredients can be added as long as the total water activity remains below 0.2 or, ideally for long shelf lives, below 0.15. The relative humidity of the processing facility must also be kept low so that the probiotic dietary supplement does not absorb moisture from the environment during production.

Once the low-water-activity probiotic format has been produced, those conditions can be maintained by choosing packaging with adequate moisture vapor transmission rates (MVTR) (Table 5). The lower the MVTR, the slower the moisture ingress. Preventing moisture ingress helps to maintain a viable probiotic cell count over the shelf life of the product.

Not all plastic bottles are equal in terms of sustaining probiotic viability. Polyethylene terephthalate (PET) bottles should never be used, as their structure allows migration of too much moisture compared with high-density polyethylene (HDPE). While glass bottles have the best MVTR, a bottle seal that adheres well to glass must be chosen to prevent moisture ingress from the bottle opening. The addition of desiccant packs into the bottles aids in maintaining the low water activity of the probiotic contents. CSP^®^ brand vials with a desiccant in the bottle wall have proven to be highly effective at maintaining the low water activity of the probiotic contents throughout shelf life, even at high-humidity storage conditions, as recommended by the International Conference on Harmonization (ICH).

Chewable tablets and mixing probiotics with other active ingredients are two trends currently seen with commercially available dietary supplements. These two products introduce additional variables which may impact probiotic count and survival. Compression during tableting tends to reduce the viability of probiotics in the tablet [33], and other active ingredients can contribute to an increase in water activity (as discussed above). Also, ingredients of plant origin may be rich in polyphenols, which may be antimicrobial [34].

The compression pressure used in chewable tablet production can decrease the probiotic cell count dramatically. The extent of the compression cell count loss will depend on the formulation and probiotic strain [35]. Tests should be performed to understand the minimum amount of compression required to produce a tablet with acceptable friability characteristics in an effort to preserve as many live cells as possible. As mentioned above, ingredients in the chewable tablet formulation should be chosen with the lowest possible water activity. Using packaging such as the CSP^®^ vial can help to maintain low water activity and probiotic cell count throughout shelf life.

When considering other active ingredients to be mixed with probiotics, there are generally two stages that should be evaluated. In the dietary supplement format, it will typically be the water activity of the other active ingredient(s) that can be detrimental to probiotic survival if they increase water activity. The second consideration is the impact of the active ingredient(s) on the freeze-dried probiotics upon rehydration, which occurs after ingestion in the stomach or when a buffer solution is added before enumeration of the product in the laboratory [16]. The exact impact of the active ingredient(s) on the probiotic in vivo can be difficult to determine because of the logistics required to sample from a person. Gastric simulators can be employed for an approximation of the in vivo system, but then careful consideration should be taken regarding simulator set up.

For interaction of the active ingredient(s) with the freeze-dried probiotic during the enumeration process, an interaction assay should be first conducted to determine if the ingredient decreases the probiotic count upon contact. The interaction assay could consist of enumerating a rehydrated sample before and after a 30-min hold time. If the cell count is reduced after 30 min, mitigation steps in the enumeration procedure, such as making a bigger initial dilution (e.g., 1:100), can reduce the impact and interaction of the active ingredient(s) with the probiotic.

## 11. Inclusion of Probiotics in Foods and Beverages

The prevalence of nondairy foods containing probiotics has steadily been on the rise, especially as consumers begin to experience “pill fatigue”. Consuming probiotics in a food may be perceived as a more natural way of receiving a daily dose [36].

The shelf life of probiotic nondairy foods is generally shorter than that of dietary supplements, but the matrices can be harsher on probiotic survival. When adding probiotics to juice, there are many factors to consider: pH, acids, anthocyanins, and the fact that the probiotic is in a vegetative rather than a freeze-dried state, as in dietary supplements [37,38]. Refrigeration of the juice is required to help maintain probiotic viability so that the adequate dose is delivered throughout shelf life and also to avoid metabolic activity of the probiotic and spoiling of the juice [39].

Secondary packaging options enable the production of shelf-stable juice, where the probiotic is in a separate compartment (e.g., in the bottle cap or a straw) and only released into the juice immediately before consumption [17].

There are many food formats that can successfully incorporate probiotics and deliver them at the required dose and usually involve experimentation to find the optimal formulation and strain combination. Chocolate, crackers, breakfast cereal, snacks, chips, peanut butter, and crispy granola bars are among the many options for probiotic foods. The main factor impacting probiotic stability in these food formats is water activity. Generally, the water activity must not be more than 0.25 to meet a 12-month shelf life at 25 °C. The exception to this water activity guideline is fat-based foods such as chocolate and peanut butter, where good probiotic stability can be achieved despite higher water activities up to 0.4.

### 11.1. Fermented Milk Products

Possibly the best-known traditional probiotic foods, yogurts and similar fermented milk products, have a strong association with gut health in many societies globally. Fermented milk products stand out as a special type of matrix for carrying probiotic health benefits, in that they offer the possibility of increasing the probiotic population during the fermentation step, thus providing cost-efficient cell counts. The downside, however, is that care must be taken to assure that growth of an added probiotic culture does not compromise the sensory profile of the product. Thus, in order to benefit from this, extra diligence must be given to quality assurance during prototype development (sensory aspects) as well as during commercial production, where the clinically relevant dose must be reliably met during manufacturing. Furthermore, if not managed carefully, the fermentation step imposes a potential health risk, in that pathogens may be allowed to grow alongside probiotic and starter cultures. Such a risk can be mitigated by applying dedicated and strict hygiene standards for fermented products.

The probiotic genera *Lactobacillus* and *Bifidobacterium*, which have a long tradition as starter cultures in yogurt production, generally exhibit good survivability in fermented milks, although very-low-pH products can be too acidic for certain strains [40]. Thus, finding the right balance between clinical dose, shelf life, and cost efficiency will in many cases be a function of strain type and pH. Other parameters that influence successful incorporation of probiotics include fermentation temperature (affecting probiotic growth), storage temperature, packaging type (oxygen transmissibility), processing steps such as heat treatment and homogenization, and interaction with other ingredients [41,42]. As the addition of fruits and (increasingly) grains is commonplace in yogurt production, it should be stressed that certain types of fruits and grains can be particularly detrimental to probiotic survivability [43]. Therefore, prescreening the compatibility of a given probiotic strain with select ingredients of interest is a recommended approach to prototype development.

Being liquid or semiliquid food applications, fermented milk products generally benefit from the given probiotic being added as a frozen pellet rather than lyophilized/milled powder. Pellets will more conveniently allow homogeneity of the product and reduce mixing time, whereas powdered probiotics can lead to issues with wettability and dispersibility.

### 11.2. Ice Cream

Ice cream poses another interesting yet different matrix for carrying probiotics. In contrast to fermented dairy, where probiotic viability is measured in weeks or, in optimal cases, months, ice cream is able to accommodate probiotic strains at clinical levels for more than a year [44]. When stored appropriately (−18 °C or less), ice cream may accommodate probiotics for longer than any other dairy application (1 year or more) due to its frozen format. Other factors typically associated with ice cream which aid probiotic stability include neutral (or closer to neutral) pH, high total solids, and, especially, fat content.

There are, however, obstacles to overcome before successful inclusion of probiotics is achieved in ice cream products. Overrun, which aerates the ice cream mix prior to packaging, may cause a drop in viable cell counts due to increased oxygen incorporation. This can be compensated for by adding overage or by culturing the probiotic population during a potential fermentation step prior to the actual ice cream manufacture. Additionally, the toxic activity of oxygen might be accommodated by utilizing aerotolerant species such as lactobacilli, rather than strictly anaerobic species. Another special challenge associated with probiotic ice cream production is the freezing step, which can compromise the bacterial cell envelope and thus impose a decrease in cell count in its own right [45]. In order to minimize this threat, it may be advisable to employ a rapid freezing step to control ice crystal formation in both the product and the bacterial cells within.

### 11.3. Probiotic Cheese

Although probiotics in cheese are not widespread, cheese is very well suited as a carrier and many examples of successful inclusion have already been documented [46]. Most cheeses have a high fat content, often combined with a relatively low water content. When further combined with cool storage conditions, this allows for probiotic viability at clinically relevant levels after several months of shelf life. The propensity for growing probiotic cell counts during the ripening process in cheese-making even makes it feasible to include a typical daily clinical dose of probiotics in very small amounts of cheese. Strains such as *Lactobacillus acidophilus* NCFM, *Lactobacillus paracasei* Lpc-37, and *Lactobacillus rhamnosus* HN001 have successfully been included in a standard gouda cheese with cell counts in excess of 10^8^ CFU/g after 200 days [46,47,48]. Thus, a mere serving of 10 g of cheese would be sufficient to obtain the desired daily dose of 10^9^ CFU [49].

### 11.4. Other Dairy Applications

Many other types of dairy applications exist and hold promise as vectors of probiotic health benefits. The general compatibility of most probiotic species with the components of milk, combined with the underlying requirement that most dairy products must be stored at low temperatures, make the two a great match and provide many avenues for the continued development of probiotics in dairy products [50].

## 12. Inclusion of Probiotics in “Medical Devices”

As described in Table 1, according to the definition, probiotics do not need to be consumed but can be administered in any other form. This opens the opportunity for so-called medical device applications. Most probiotic medical devices fall in Class I, meaning they are not meant to help support or sustain life or be substantially important in preventing impairment to human health and may not present an unreasonable risk of illness or injury [51]. In the European Union, however, probiotics, as living microorganisms, are specifically excluded from the medical device regulation [52].

Hence, lotions, mouth washes, and vagitories can be considered for medical devices. A common application of probiotics in a medical device is for vaginal application; these are often vagitories [53] but may also be tampons, etc. The probiotics included in such products can be different from the species that are used for oral administration, but they do not have to be. The production process of such probiotics is not different from other probiotics in the sense that it needs to be specially adapted to the medical device delivery format. As with all probiotic strains, production will be strain dependent.

The technological requirements of the strains are different in medical devices as compared with foods or dietary supplements. However, the basic challenges for maintaining viability are the same: water activity, oxygen, temperature, pH, etc. Shelf life is usually similar to that of dietary supplements: 24 months at room temperature.

## 13. Conclusions

Once human intervention studies have documented that microorganisms qualify as probiotics, the next step will need to be taken to see if the strains can be cultured at an industrial scale and if they can be successfully incorporated in consumer products. Preferably, this part of probiotic commercialization will run in parallel with clinical trials, to avoid studying an uncommercializable strain. Culturing at a large scale and industrial processing set very different requirements for strains than laboratory scale culturing; also, medium requirements are very different due to cost and other factors. To ensure the consistent high quality of the strains, a quality control program needs to be established to assure the consistent quality of everything, from ingredients to final product, and a quality assurance program needs to be in place to run reliable production processes. This all requires rigorous documentation of procedures and results. Once high-quality probiotic bulk has been produced, the strain(s) need to be incorporated into consumer products. These products have different requirements, from shelf life to storage conditions and product composition. In any case, a minimal efficacious dose should be delivered to the consumer at the end of shelf life. Probiotics, being live microorganisms, make this all challenging. However, by choosing the right strains, culture conditions, and product manufacturing, much can be achieved and the investigated health benefits can be delivered to the consumer.

## Figures and Tables

**Figure 1 microorganisms-07-00083-f001:**
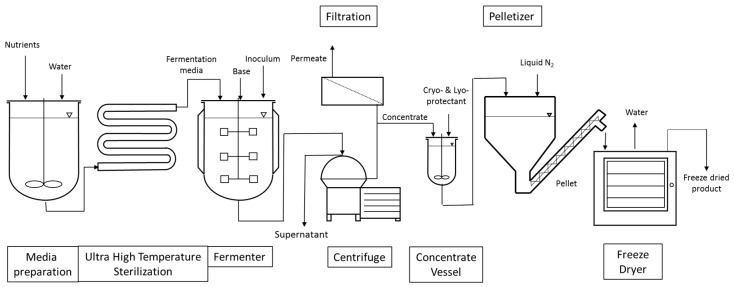
Schematic representation of the production of probiotics for dietary supplements and dairy starter culture strains.

**Table 1 microorganisms-07-00083-t001:** Implications of the probiotic definition; set forth by Hill et al. [2].

1	are microorganisms	Although most commercial probiotics are lactobacilli and bifidobacteria, they can be other microbes and do not need to be bacteria.
2	need to be alive	When administered; while it may be desirable that they are alive in the gastrointestinal tract, it is not required.
3	need to be administered	This does not imply they must be eaten; other routes of administration are possible.
4	in sufficient amounts	At the end of shelf life, there are still at least as many viable microbes in the product as were used in a clinical study.
5	need to have a health benefit	This benefit should be shown in the target host population.

**Table 2 microorganisms-07-00083-t002:** Examples of regulatory guidelines for quality control.

1	21 CFR 117: Code of Federal Regulations	Current Good Manufacturing Practice, Hazard Analysis, and Risk-Based Preventive Controls For Human Food
2	21 CFR 111: Code of Federal Regulations	Current Good Manufacturing Practice In Manufacturing, Packaging, Labeling, Or Holding Operations For Dietary Supplements
3	ICH Q7: International Council for Harmonization of Technical Requirements for Pharmaceuticals for Human Use	Good Manufacturing Practice Guide For Active Pharmaceutical Ingredients

**Table 3 microorganisms-07-00083-t003:** Steps required for managing off-specification materials.

1	Identifying nonconforming products or materials at any stage of the process
2	Investigating nonconforming products to provide critical information and determine corrective and preventative actions
3	Isolating nonconforming products to prevent unintended use
4	Notifying all affected departments of the nonconformance
5	Determination of the disposition

**Table 4 microorganisms-07-00083-t004:** Examples of end-product testing.

1	Physical examination
2	Functionality—the need to prove that the product will do what the customer expects. For probiotic bacteria, this is proven by establishing a label claim and meeting that claim per each batch produced.
3	Absence of pathogens
4	Cross contamination/hygiene—more and more customers want minimal cross contamination and hygiene issues such as yeast/mold absent.
5	Identification has progressed from microscopy and phenotypic traits that offered high-level distinction of probiotics to more specific genotypic assays by means of PCR technologies. Riboprinting and 16S sequencing have been an industry standard of probiotic identification in recent years. These genetically based methods distinguish at the species level and, in some cases, beyond. Species of bacteria are often made up of several different strains and probiotics are typically sold by strain designation. One definition of a strain is a difference of at least one base pair in a bacteria’s genome. Strain-specific PCR assays can be designed to target sequences of a genome which uniquely identify the bacteria (probiotic) at the strain level. The importance of these assays become more apparent, not only as new research reveals the difference a single base pair change can have, but also as the dynamics of the probiotic industry continue to evolve.

**Table 5 microorganisms-07-00083-t005:** Examples of moisture vapor transmission rates (MVTR) for commonly used bottle types for probiotic dietary supplement products (from the Alpha Packaging Plastics Comparison Chart (http://www.alphap.com/bottle-basics/plastics-comparison-chart.php)).

Material	MVTR (g*mil/100 in^2^/24 h)
PET (Oriented or Stretch-Blown Polyethylene Terephthalate)	2.0
HDPE (High-Density Polyethylene)	0.5
Glass	near 0

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
