# Peer review of "The Production and Delivery of Probiotics: A Review of a Practical Approach"

_microorganisms, 2019, doi:10.3390/microorganisms7030083_

Round 1
Reviewer 1 Report
Figure 1 is blurry, can be improved.
Throughout, the author have most mentioned about lactobacillus or bifido, breif addition of other strains and their information can be beneficial
In the market, most of the probiotic products are sold as fridge-free-technology. Adding a brieft comment on that aspect can be beneficial.
Adding to the point 3, importance of temperatures for production and/or stability on the efficacy is missing.
No recent references were used.
Author Response
Figure 1 is blurry, can be improved
We have included a higher resolution figure
Throughout, the author have most mentioned about lactobacillus or bifido, breif addition of other strains and their information can be beneficial
Correct, we have mainly focused on bifidobacteria and lactobacilli. We now have mentioned the use of strains from other (also more robust) genera, in the manuscript
In the market, most of the probiotic products are sold as fridge-free-technology. Adding a brieft comment on that aspect can be beneficial.
Correct, most product are, actually sold at ambient temperatures, we will comment this
Adding to the point 3, importance of temperatures for production and/or stability on the efficacy is missing.
Indeed, processing temperatures (and humidity) are main factors affecting stability and thereby also efficacy. This has been added to the manuscript.
No recent references were used.
This has been pointed out also by other reviewers and amended.
Reviewer 2 Report
The review is very interesting and important for readers involved in probiotics are. I hope that considering of the following remarks will improve the quality of this very good prepared review.
line 39 the problem of adequate amount of probiotics was discussed in the following publication, cited by authors: Hill, C. et al. The International Scientific Association for Probiotics and Prebiotics consensus statement on the scope and appropriate use of the term probioticNat. Rev. Gastroenterol. Hepatol. 2014;11:506–514 and should be shortly described in the publication. Please add also short explanation of QPS issue
line 73 please inform readers more about cryoprotectants
line 102 – Fig. 1 – please add detailed description of this important figure, please explain all abbreviations, for readers would be useful to add for every process the requirements concerning temperature and humidity and purity classes in the different industrial areas.
line 310 – please discuss the very important problem of CFU in case of multistrain probiotics. Usually producers inform only about for example total lactic bacteria CFU. Are there other methods of determination of alive bacteria in multistrain products?
line 326 – the table with MVTR of different materials (foils, etc.) would be very useful for readers
line 350 – could you expand the description of a rehydration process, which can occur in water before ingestion and also in the intestine?
Line 430 – could you shortly discuss the problem of probiotics in sweets like chocolate for example?
Author Response
The review is very interesting and important for readers involved in probiotics are. I hope that considering of the following remarks will improve the quality of this very good prepared review.
line 39 the problem of adequate amount of probiotics was discussed in the following publication, cited by authors: Hill, C. et al. The International Scientific Association for Probiotics and Prebiotics consensus statement on the scope and appropriate use of the term probioticNat. Rev. Gastroenterol. Hepatol. 2014;11:506–514 and should be shortly described in the publication. Please add also short explanation of QPS issue
A comment has been added on dose, referring to Hill et al. 2014, as suggested.
We have not added an explanation on QPS as we feel that safety falls outside the scope of the manuscript. Furthermore, if we included QPS, we would also need to comment on GRAS, etc.
line 73 please inform readers more about cryoprotectants
Cryoprotectants and lyoprotectants have been further explained and a relevant reference included.
line 102 – Fig. 1 – please add detailed description of this important figure, please explain all abbreviations, for readers would be useful to add for every process the requirements concerning temperature and humidity and purity classes in the different industrial areas.
We have redrawn the figure and written everything in full so it is easier to understand for readers less familiar with process.
line 310 – please discuss the very important problem of CFU in case of multistrain probiotics. Usually producers inform only about for example total lactic bacteria CFU. Are there other methods of determination of alive bacteria in multistrain products?
It remains challenging to enumerate the separate strains in a multi-strain product. Experimental methods exist. This has been included in the text.
line 326 – the table with MVTR of different materials (foils, etc.) would be very useful for readers
A table has been included as suggested
line 350 – could you expand the description of a rehydration process, which can occur in water before ingestion and also in the intestine?
A paragraph has been added on this topic
Line 430 – could you shortly discuss the problem of probiotics in sweets like chocolate for example?
The paragraph on confectionary; including chocolate and peanut butter, as carrier for probiotics has been expanded as suggested
Reviewer 3 Report
The manuscript is well written on the important and practical topics relevant to the production and commercialization of probiotics.
The only thing that need to be improved is include more sufficient references for each topics. There is almost no reference from topic 2 to 10.1.
Once again, the authors should include more references to the manuscript for more comprehensive understanding of each topic.
Author Response
The manuscript is well written on the important and practical topics relevant to the production and commercialization of probiotics.
The only thing that need to be improved is include more sufficient references for each topics. There is almost no reference from topic 2 to 10.1.
This has also been pointed out by the other reviewers; more and up to date references have been included.
Once again, the authors should include more references to the manuscript for more comprehensive understanding of each topic.
This has also been pointed out by the other reviewers; more and up to date references have been included.
Reviewer 4 Report
BROAD COMMENTS
Brief summary
In general, the aim of this work was to discuss the incorporation of probiotic bacteria into food systems through different production processes.
However, at the end of abstract it should be mentioned the main overview of what this paper gives.
On the other hand, a weakness could be reported in the fact that too few literature sources are provided. This paper may address to a specific approach but is still a review article, otherwise is a Report of a practical approach.
SPESIFIC COMMENTS
Line 44, 45: Please specify providing the appropriate literature.
Line 62: Please correct ‘’Figure 1’’ to (Figure 1) or ‘’as shown in Figure 1’’.
Chapter 3 do not provide any references. The statements that are mentioned about nutritional requirements for specific Lactobacilli has to be ‘validated’ by corresponding references.
Line 128-131: It should be proper to add some reference for this statement?
Line 136-135: Any reports of the affected performance?
Line 148: What is Six Sigma referring to?
Please check font on line 220, also 242-248
Line 320: Ref. required
Line 339-341: Ref. required
Line 355: Any Ref?
Line 360-361: How is this supported by literature studies?
Line 363-365: Any Ref?
Line 386-388: Ref. required
Line 398: Any Ref?
Line 426-428: Ref required
Author Response
In general, the aim of this work was to discuss the incorporation of probiotic bacteria into food systems through different production processes.
However, at the end of abstract it should be mentioned the main overview of what this paper gives.
On the other hand, a weakness could be reported in the fact that too few literature sources are provided. This paper may address to a specific approach but is still a review article, otherwise is a Report of a practical approach.
Due to the word limit we have included the aim of the paper in the beginning of the abstract
The reviewer is correct that the paper included too few references, this was also pointed out by the other reviewers and has been amended.
SPESIFIC COMMENTS
Line 44, 45: Please specify providing the appropriate literature.
Reference has been included
Line 62: Please correct ‘’Figure 1’’ to (Figure 1) or ‘’as shown in Figure 1’’.
Corrected as suggested
Chapter 3 do not provide any references. The statements that are mentioned about nutritional requirements for specific Lactobacilli has to be ‘validated’ by corresponding references.
References added
Line 128-131: It should be proper to add some reference for this statement?
The section expanded to clarify and a reference added.
Line 136-135: Any reports of the affected performance?
References added
Line 148: What is Six Sigma referring to?
References added
Please check font on line 220, also 242-248
Corrected
Line 320: Ref. required
References added
Line 339-341: Ref. required
Section expanded and references added
Line 355: Any Ref?
There is no reference for this, this is based on experience and indicated as such in the text.
Line 360-361: How is this supported by literature studies?
Reference added
Line 363-365: Any Ref?
Reference added
Line 386-388: Ref. required
Reference added
Line 426-428: Ref required
Reference added
Reviewer 5 Report
The paper is well written, but it would be interesting to add additional information
1. A brief note on Probiotic demand and how it expanded in global market and more details on how it supplements/improve health condition is recommend in Section 1
2. What does author mean by “Preferably at least the dose that was shown to be efficacious; and at end of shelf-life” reframe the sentence for better understanding of the readers.
3. Author needs to elaborate on most challenges in the production of probiotics like scale up in manufacturing or loss of cells during freeze drying
4. Discuss on the advance tools used to understand the nutritional requirements of the probiotics,
5. A brief outline on advance technologies used in strain improvement in developing a robust strain is recommended in Section 6.
6. It would be good to discuss characteristics of probiotics such as antimicrobial, Anticarcinogenic properties, Immunologic enhancement etc.
Author Response
The paper is well written, but it would be interesting to add additional information
1. A brief note on Probiotic demand and how it expanded in global market and more details on how it supplements/improve health condition is recommend in Section 1
Although the commercial and health benefit aspects fall outside the scope of the review, we have included a short paragraph on this.
2. What does author mean by “Preferably at least the dose that was shown to be efficacious; and at end of shelf-life” reframe the sentence for better understanding of the readers.
We mean that at the end of shelf-life there are still at least as many viable microbes in the product as were used in a clinical study. We have rephrased the sentence to clarify this.
3. Author needs to elaborate on most challenges in the production of probiotics like scale up in manufacturing or loss of cells during freeze drying
A new section on scaling up, including freeze-drying has been added.
4. Discuss on the advance tools used to understand the nutritional requirements of the probiotics,
A paragraph on this topic has been added
5. A brief outline on advance technologies used in strain improvement in developing a robust strain is recommended in Section 6.
This has been included in the nutritional requirements paragraph as requested above
6. It would be good to discuss characteristics of probiotics such as antimicrobial, Anticarcinogenic properties, Immunologic enhancement etc.
The specific health benefits fall outside the scope of the review. We have therefore included a small section in the introduction to direct the reader to these and other aspects of probiotics.